# Lipid Metabolism and Cancer Immunotherapy: Immunosuppressive Myeloid Cells at the Crossroad

**DOI:** 10.3390/ijms21165845

**Published:** 2020-08-14

**Authors:** Augusto Bleve, Barbara Durante, Antonio Sica, Francesca Maria Consonni

**Affiliations:** 1Department of Pharmaceutical Sciences, Università del Piemonte Orientale “Amedeo Avogadro”, Largo Donegani, 2-28100 Novara, Italy; augusto.bleve@uniupo.it (A.B.); barbara.durante@humanitasresearch.it (B.D.); francesca.consonni@uniupo.it (F.M.C.); 2Humanitas Clinical and Research Center–IRCCS–, via Manzoni 56, Rozzano, 20089 Milan, Italy

**Keywords:** tumor-associated macrophages (TAMs), myeloid-derived suppressor cells (MDSCs), cancer immunotherapy, lipid metabolism, obesity, fatty acids, cholesterol

## Abstract

Cancer progression generates a chronic inflammatory state that dramatically influences hematopoiesis, originating different subsets of immune cells that can exert pro- or anti-tumor roles. Commitment towards one of these opposing phenotypes is driven by inflammatory and metabolic stimuli derived from the tumor-microenvironment (TME). Current immunotherapy protocols are based on the reprogramming of both specific and innate immune responses, in order to boost the intrinsic anti-tumoral activity of both compartments. Growing pre-clinical and clinical evidence highlights the key role of metabolism as a major influence on both immune and clinical responses of cancer patients. Indeed, nutrient competition (i.e., amino acids, glucose, fatty acids) between proliferating cancer cells and immune cells, together with inflammatory mediators, drastically affect the functionality of innate and adaptive immune cells, as well as their functional cross-talk. This review discusses new advances on the complex interplay between cancer-related inflammation, myeloid cell differentiation and lipid metabolism, highlighting the therapeutic potential of metabolic interventions as modulators of anticancer immune responses and catalysts of anticancer immunotherapy.

## 1. Introduction

Hematopoiesis consists of a rigorous series of cell lineage commitments that regulate the differentiation of hematopoietic stem cells (HSCs) into lymphoid and myeloid progenitors and subsequently to mature immune cells necessary to maintain the physiological levels of circulating leukocytes [1]. Under conditions of immunological stress (e.g., infection and cancer) the signals arriving from damaged tissues create an “emergency” hematopoiesis, which guarantees an increase in the supply of lymphoid and myeloid cells [2]. In particular, cancer growth drives the expansion of immunosuppressive and tumor-promoting myeloid populations, mainly tumor-associated macrophages (TAMs) and myeloid-derived suppressor cells (MDSCs) [3].

“Emergency” myelopoiesis is a highly coordinated process orchestrated by cytokines and growth factors, produced by cancer cells and other cellular components of the tumor microenvironment (TME) [4,5]. Despite some degree of functional overlap, chemokines and complement components (i.e., CCL2, C5a) are specialized determinants of macrophage recruitment in tumors, while inflammatory cytokines (i.e., IL-1β and IL-6) and myeloid growth factors (i.e., M-CSF, GM-CSF, G-CSF) critically orchestrate emergency myelopoiesis [6,7]. The latter act through activation of specific transcription factors that differentially drive terminal maturation of immune cells [7]. In particular, interleukin-1β (IL-1β) has been found to increase the proliferation and differentiation rate of HSCs through upregulation of PU.1 transcription factor. Whereas CCAAT-enhancer-binding protein (C/EBP)α appears to be a major regulator of “steady-state” granulopoiesis, C/EBPβ and Signal Transducer and Activator of Transcription 3 (STAT3) promote expansion and maturation of neutrophils in emergency conditions. Moreover, IL-17A promotes both granulocyte colony-stimulating factor (G-CSF)- and stem cell factor-mediated neutrophilia, and supports G-CSF-driven emergency myelopoiesis [8]. Terminal differentiation of macrophages is instead induced by macrophage-CSF (M-CSF or CSF1), through activation of interferon regulatory factor 8 (IRF8) and PU.1, which guide commitment of myeloid progenitors towards the monocytic/macrophage lineage [9]. Retinoic acid-related orphan receptor γ (RORC1/RORγ) orchestrates emergency myelopoiesis by promoting C/EBPβ, IRF8 and PU.1 transcription factor [10].

TAMs and MDSCs are essential components of the immunosuppressive TME, which promotes tumor evasion and unresponsiveness to conventional chemo- and radio-therapy [11,12,13,14]. Cancer immunotherapy has emerged as revolutionary therapeutic approach, potentially able to overcome the suppressive TME [15,16]. Immune checkpoint blockade (e.g., anti-PD1/PD-L1), adoptive cell transfer (e.g., CAR T), cancer vaccines (e.g., Sipuleucel-T) and immunostimulatory cytokines (e.g., IL-2) are the principal immunotherapeutic tools [16,17,18]. Nevertheless, a substantial number of patients result unresponsive to these treatments, a clinical result strongly and negatively influenced by the expansion of suppressive myeloid cells [16].

Increasing evidence suggests that energy metabolism could be responsible for the failure of antitumor immunity [19,20]. Beyond the influence of inflammatory mediators released during cancer progression, metabolic adaption and nutrient competition for essential nutrients (e.g., amino acids, glucose, fatty acids, oxygen) between proliferating cancer cells and tumor-infiltrating immune cells drastically modify their metabolic state and functional phenotypes [21,22,23].

Of relevance, the diverse polarization states of macrophages (M1 vs M2) are characterized by distinct patterns of lipid metabolism (fatty acids synthesis (FAS) vs fatty acids oxidation (FAO) respectively) [24,25]. Further, the systemic deregulation of lipid metabolism in obese subjects appears increasingly relevant in modulating cancer-related inflammation [26], expansion of myeloid cells and inflammatory phenotypes [27,28].

Within this scenario, therapeutic interventions modulating key molecular players of lipid metabolism appear promising tools for the antitumor reprogramming of TAMs and MDSCs, able to restore effective anti-tumor immunity. Here, we discuss the complex interplay between cancer-related inflammation, myeloid cell differentiation and lipid metabolism, highlighting the therapeutic potential of interventions in lipid metabolism as modulators of anticancer immune responses and catalysts of anticancer immunotherapy.

## 2. Tumor-Associated Myeloid Cells

Immunological stresses, such as infection or cancer, shift the steady-state hematopoiesis toward ‘emergency myelopoiesis’, to cope with the increased demand for granulocytes and monocytes [3,14]. Cancers drive the expansion of heterogeneous immature myeloid cells [8,9] through elicitation of tumor-derived factors (TDFs), including growth stimulating factors (e.g., CSFs), cytokines (e.g., IL-10, IL-6, IL-4, IL-8) and prostaglandins; these act in a paracrine and/or systemic fashion to sculpt a pro-tumor microenvironment [29] and to guide the pathological differentiation of myeloid lineage into alternatively activated and immunosuppressive cells, mainly represented by tumor-associated macrophages (TAMs) and myeloid-derived suppressor cells (MDSCs) [30].

### 2.1. Tumor-Associated Macrophages (TAMs)

Macrophages are essential tissue sensors with the ability to orchestrate articulated programs in order to scan pathogenic insults, generate pro-inflammatory signals, promote the resolution of inflammatory response and restore tissue homeostasis. This remarkable functional plasticity is hijacked in diseases where macrophages act as essential pathogenic promoters.

In response to microenvironmental signals, macrophages develop a classical (M1) or alternative (M2) activation state, respectively, owning anti-tumor or pro-tumorigenic properties [31]. M1/M2 polarization of macrophages is a simplistic, although useful, concept that emphasizes the extremes of a continuum depicting macrophage plasticity [32]. Clinical and experimental data indicate that, in the large majority of cancers, TAMs acquire an M2-like tumor-promoting phenotype characterized by high levels of immunosuppressive markers (e.g., ARG1, MMR1, IL10), as opposed to low levels of inflammatory cytokines (e.g., IL-12, IL1β, TNFα) [32,33,34]. However, different tumor types and/or disease stages generate specific inflammatory microenvironments, which drive the remarkable heterogeneity of TAMs [34,35,36]. TAMs are able to promote cancer progression through: promotion of tumor angiogenesis, tumor cell intravasation and metastasis, and suppression of adaptive and innate anti-tumor immunity [37].

TAMs frequently accumulate in hypoxic regions and produce pro-angiogenic factors and cytokines (e.g., VEGF, PDGF, TGFβ), which promote neovascularization and vascular permeability, favoring cancer cells growth and intravasation [37,38]. Further, a specific subset of TAMs expressing the angiopoietin receptor Tie2 resides in close contact with tumor blood vessels, providing pro-angiogenic factors [38,39]. TAMs also contribute to the remodelling of the extracellular matrix (ECM) through the secretion of multiple proteases, such as matrix metalloproteinases (MMPs) and serine/cysteine proteases and, thus, facilitating the spread of distal tumor cells [40]. Further, TAMs express lymphangiogenic factors, namely VEGF-C that promotes tumor lymph angiogenesis and lymphatic metastasis [41].

TAMs can abate T cells’ antitumor responses by secreting anti-inflammatory cytokines (i.e., TGFβ, IL-10) and chemokines (i.e., CCL22, CCL17), which recruit Tregs and T helper (Th)2 cells [32], expressing PD-L1 and CD80 ligands for the inhibitory T cell receptors PD-1 and CTLA4, respectively, and by reducing the bioavailability of L-arginine, an essential metabolite for T cell functions [42,43]. TAMs also induce γδ-T cells to secrete IL-17 that orchestrates a pro-metastatic response mediated by neutrophils [44]. A recent clinical study identified CCL8 as an additional pro-tumorigenic TAMs effector molecule that induces the expression of an invasive gene signature in cancer cells [45]. Although TAMs are largely immunosuppressive, specific subsets may also promote anti-tumor immunity [11,31]. These M1-like TAMs produce IFNγ, express high levels of both major histocompatibility complex (MHC)-II and costimulatory molecules and secrete chemokines and cytokines (e.g., CXCL9, CXCL10, IL-18, IL-22) that boost Th1 and NK cells’ pro-inflammatory and anti-tumoral activity [43,46,47]. In this view, re-educating TAMs to acquire anti-tumor functions represents an attractive strategy to alleviate the immunosuppressive features of TME and to promote tumor regression [17].

Although TAMs were considered to arise predominantly from bone marrow-derived monocytes [48], in different cancer types, tissue-resident macrophages (TRMs), originally derived from yolk sac, were reported to contribute to the pool of TAMs directly supporting cancer cell proliferation and tumor progression [7,49,50,51,52]. In an inducible lung carcinoma model, a percentage of TAMs were showed to arise from spleen-derived monocytes [53]. Thus, the different origins of TAMs in distinct cancer types may affect tumor progression and responsiveness to anti-cancer treatments.

### 2.2. Myeloid-Derived Suppressor Cells (MDSCs)

MDSCs are a group of highly heterogenous immature myeloid cells conventionally divided into two major subpopulations: polymorphonuclear (PMN)-MDSCs and monocytic (M)-MDSCs [54]. Phenotypically, murine PMN-MDSCs are characterized as CD11b^+^Ly6C^low^Ly6G^+^, while M-MDSCs are CD11b^+^Ly6C^hi^Ly6G^−^. Human PMN-MDSCs are instead defined as HLA-DR^low/−^CD14^−^CD11b^+^CD15^+^, while M-MDSCs as CD11b^+^CD14^+^HLA-DR^low/−^CD15^−^ [55]. However, it is very challenging to discriminate immunosuppressive PMN- and M-MDSCs from their pro-inflammatory counterparts (neutrophils and inflammatory monocytes, respectively) as they share cellular origin and phenotypic markers [55].

In cancer bearers, MDSCs’ occurrence is governed by a network of transcriptional regulators promoting immature and immunosuppressive activation of myeloid cells. STAT3 was the first transcription factor characterized for its capacity to drive MDSCs expansion and accumulation [56]. NF-κB and JAK/STAT signalling are known to upregulate immunosuppressive mediators such as inducible nitric oxide synthase (iNOS) (in M-MDSCs), reactive oxygen species (ROS) and arginase 1 (ARG1) (in PMN-MDSCs) [57]. Other studies have shown that C/EBPβ promotes granulocytic cells’ expansion in emergency conditions [58], while the RORC1/RORγ transcription factor orchestrates emergency myelopoiesis in response to TDFs (i.e., G- and GM-CSF) [10]. Conversely, IRF8 acts as a negative regulator of MDSCs generation, due to its capacity to trigger the terminal differentiation into mature and pro-inflammatory myeloid cells [59]. The NLRP3 inflammasome, producing IL-1β and IL-18, is up-regulated in MDSCs and its lack decreased MDSCs frequency [60]. Several cytokines and chemokines (e.g., GM-CSF, G-CSF, CCL2, CXCL1) induce the mobilization of MDSCs from bone marrow to the peripheral lymphoid organs and to the TME, where they promote tumor immune evasion through direct or indirect mechanisms, both antigen-specific and unspecific [61].

Specific amino acids’ depletion is one of the main mechanisms by which MDSCs inhibit anti-tumoral T cell activity: L-arginine depletion by ARG1 mediates the downregulation of T cell receptor (TCR)-ζ chain expression and inhibition of cyclin-dependent kinases (CDKs) pathway regulating cell cycle [62]; tryptophan catabolism into kynurenines by indoleamine 2,3-dioxygenase (IDO) induces T cell cycle arrest and the differentiation of immunosuppressive Tregs (via Foxp3 induction) [63,64]; absence of the alanine-serine-cysteine (ASC) exporter on MDSCs restrains the extracellular pool of cysteine strictly required for T cell activation [65].

Several TDFs, such as GM-CSF, VEGF, TGFβ, IL-6 and IL-10, stimulate MDSCs to overproduce ROS and RNS, which induce TCR-ζ chain downregulation/alteration (i.e., nitration/nitrosylation), impairing TCR signalling and T cell activation [66]. Tumor-derived prostaglandin E2 (PGE2) induces the overexpression of iNOS in M-MDSCs (via p50-NF-κB signalling), thus enhancing the production of nitric oxide (NO), able to reduce MHC-II expression and induce T-cell apoptosis [67]. In addition, hypoxia (via HIF-1α), IFNγ (via STAT1-IRF1 axis), as well as M-CSF, VEGF-A and cyclooxygenase (COX)2/PGE2 axis were reported to induce PD-L1 expression in MDSCs, hampering T cell anti-tumor responses [68]. MDSCs express high levels of both TGFβ, which induces Tregs differentiation and inhibits cytotoxic NK cells, and IL-10, which skews TAMs toward an M2 phenotype [8,69].

MDSCs also harbour tumor-promoting functions by non-immunological mechanisms, such as the promotion of angiogenesis (via VEGF and angiopoietins production) and matrix remodelling (via MMP9 expression), thus enabling cancer cells to move from the primary tumor to metastatic sites [61]. CXCL1, CXCL2, and CXCL5 have been shown to recruit MDSCs to the pre-metastatic niche through the interaction with CXCR2; once in the pre-metastatic site, MDSCs favor tumor cell recruitment and seeding by secreting TNFα, TGFβ, IL-6, CCL2 and CXCL2. Pro-inflammatory proteins S100A8/A9 are potent chemoattractants for MDSCs and have been implicated in the promotion of metastasis [70].

MDSCs occurrence was shown in different human cancer tissues and high frequencies of circulating MDSCs were detected in patients with different cancer types in advanced disease stages (III-IV) and associated with poor prognosis and responsiveness to radio-, chemo-, and immuno-therapy [12,13]. In sum, MDSCs represent promising biomarkers for treatment efficacy and a promising target in anticancer therapy.

## 3. Cancer Immunotherapy

The metabolic and immune heterogeneity of the tumor microenvironment facilitates tumor evasion and complicates the responsiveness to cancer treatments [71]. In this scenario, cancer immunotherapy has emerged as a revolutionary therapeutic approach, aiming at improving anti-tumor immune responses with fewer off-target effects than chemo- and radiotherapy [15,72]. Blockade of immune checkpoint molecules, adoptive T cell transfer, immunostimulatory cytokines, cancer vaccines and targeting of specific immune cell subsets are among the main immunotherapeutic approaches used to reactivate the immune system’s ability to recognize and attack cancer cells (Figure 1) [15].

### 3.1. Immune Checkpoint Blockade

Physiologically, immune checkpoints function to prevent autoimmunity or excessive immune responses, providing negative signals that restrict T cell activation. Tumor cells exploit this mechanism by deactivating tumor-infiltrating lymphocytes (TILs). In fact, activated T cells express the programmed death protein 1 (PD-1) and recognize the negative PD-1 ligand (PD-L1) signal present on the surface of cancerous cells and immunosuppressive myeloid cells. In this way, tumors escape immunosurveillance and, in concert with MDSCs and TAMs, dampen T cell activation and promote their apoptosis [73]. Therefore, blocking this interaction with specific monoclonal antibodies, defined immune checkpoint inhibitors (ICIs), restores T cell-mediated anti-tumor activity.

Cytotoxic T-lymphocyte antigen 4 (CTLA4) is a B7/CD28 family member that regulates the extent of T cell activation. It is constitutively expressed by Tregs but can also be upregulated by other T cell subsets upon activation, especially in cancer. CTLA4 competes with CD28 receptors for the binding to B7 ligands (CD80 and CD86) on antigen-presenting cells (macrophages, DCs and B cells), as well as TAMs and MDSCs, inhibiting T cell activity and thus promoting tumor progression [74]. By blocking the CTLA4/ligands, interaction T cells remain active, thus being able to recognize and kill tumor cells [75].

To date, ICIs, including PD-1/PD-L1 and CTLA4 inhibitors, represent the main class of immunotherapeutics [16,76]. Their clinical impact has grown considerably over the last decade and a large number of trials (>700 trials) involving ICIs in combination with other therapeutic approaches are ongoing [76]. However, the risk/benefit balance of their application is under critical review, due to severe side effects in numerous organs [77]. Additional immune checkpoint inhibitors have been identified, such as TIM3, TIGIT, LAG3 on T cells, and VISTA on myeloid cells, are under development and might represent alternative strategies to bypass the side effects of current ICIs [78].

### 3.2. Adoptive Cell Transfer

Adoptive cell transfer (ACT) is a treatment that uses a cancer patient’s own T lymphocytes from peripheral blood, activated and expanded ex vivo, and reinfused into patients pre-treated with lymphodepleting agents (e.g., fludarabine/cyclophosphamide), often in combination with appropriate growth factors stimulating their survival and expansion in vivo (i.e., IL-2) [79]. The most relevant types of ACT are tumor-infiltrating lymphocytes (TILs), T cells engineered for T cell receptor (TCR T) and chimeric antigen receptor T cells (CAR T) [80]. Additionally, genetic modification of NK cells is now providing promising perspectives for cancer treatment [81]. In the CAR T cell approach, peripheral blood T cells are genetically engineered to overexpress a chimeric TCR that recognizes a tumor-specific antigen in an MHC-independent manner, bypassing antigen presentation by APCs and, simultaneously, supplying the interaction with the co-stimulatory signal (e.g., CD28, CD3ζ) [82]. TCR T cell therapy, instead, consists in the overexpression of specific TCR recognizing cancer type-specific antigens (e.g., cancer–testis antigen) or patient-specific neoantigens, but unlike MHC-independent CAR T, this approach requires MHC-matching with the patient [83]. Currently, two CD19-targeting CAR T cell therapies are approved for clinical use by the Food and Drug Administration (FDA) and the European Medicines Agency (EMA) for diffuse large B cell lymphoma [84,85]. Several other CARs for tumor antigens, such as the glycolipid disialoganglioside GD2 on neuroblastoma [86] and prostate stem cell antigen (PSCA) [87], have shown strong efficacy in pre-clinical models; however, their clinical translation is still under evaluation. In addition, both CAR T and TCR T cells can cause cytokine release syndrome (CRS) and neurotoxicity, and their application for solid tumors has been challenging [88,89]. Recently, Klichinsky et al. genetically engineered human macrophages, exploiting their capacity to efficiently penetrate tumors, with CARs to direct antigen-specific phagocytic activity against tumors. In humanized mouse models, CAR macrophages (CAR-Ms) were shown to induce an M1 pro inflammatory tumor microenvironment and boost anti tumor T cell activity [90].

### 3.3. Cancer Vaccines

Cancer vaccines aim to boost the activity of tumor antigen-specific cytotoxic T cells, as well as to create a long-lasting immune memory against tumors [91]. Types of cancer vaccine include tumor cell lysates, cancer cells engineered for cytokine production, specific tumor antigens or APCs pulsed with tumor lysates [92]. Dendritic cell (DC) vaccines represent the most characterized class of cell-based cancer vaccine [93]. The approved Sipuleucel-T for the treatment of metastatic prostate cancer is generated by challenging ex vivo DCs with recombinant prostatic acid phosphatase (PAP) antigen fused with GM-CSF, which enhances antigen presentation and activation of T cells against tumor cells [93]. The cancer vaccine based on direct administration of melanoma-associated antigen-A3 (MAGE-A3), a member of tumor-specific antigens expressed by various cancerous cells, is giving promising results in the treatment of resected stage IB/II non-small cell lung cancer [94]. Interestingly, emerging clinical studies on the combination of cancer vaccines (DC- or peptide-based) with ICIs (anti-CTLA4 or anti-PD-1) have shown encouraging results in melanoma patients [95,96].

### 3.4. Cytokines

Cytokines are the first class of immunotherapy introduced in clinical practice and, unlike ICIs, can directly stimulate the growth and activity of immune cells [97]. Three main types of cytokines have been pursued for immunotherapy: interferons, which elicit antiproliferative activity [98] and induce the maturation of numerous immune cells including macrophages, NK cells, DCs, and T cells [99]; interleukins (i.e., IL-2), which elicit activity of CD4 and CD8 T cells [100]; GM-CSF, which improves T cells’ survival and supports DCs maturation [101]. Additionally, small-molecule agonists of TLR7/TLR8 directly activate maturation of APCs and promote anti-tumor activity [102]; stimulator of interferon genes (STING) agonists induce pro-inflammatory cytokine production and type I IFN-related responses, which in turn activate DCs and TAMs, promoting T cells’ priming in lymph nodes and recruitment into TME [103]. STING agonists, as single agent, as well as in combination with ICIs and cancer vaccine, showed enhanced T cell-mediated anti-tumor immunity, reduction in metastatic burden and eradication of ICI-resistant tumors in a pre-clinical model [104]. IFNα and IL-2 represent the first recombinant cytokines approved for cancer immunotherapy [105,106]; however, others, such as IL-7 and IL-15, are under clinical investigation [107,108]. Several adverse effects, such as CRS, vascular discharge and autoimmune attacks, have been observed in response to exogenous administration of cytokines and their combination with other agents is currently being evaluated to prevent such toxicity [106].

### 3.5. Reprogramming Myeloid Cells

As mentioned above, high frequency of myeloid cells in cancer patients correlates with tumor progression, metastasis formation and recurrence in many types of human tumors [54,109]. In addition, the resistance to immunotherapeutic strategies, observed in a large number of patients, can be mediated, at least in part, by the vicious cycle of inflammation and immunosuppression supported into the TME by MDSCs and TAMs [16]. Therefore, in the last decade immunotherapeutic options aiming to block or reprogram TAMs’ and/or MDSCs’ immunosuppressive activities have been explored [17,18].

#### 3.5.1. Targeting TAMs in Cancer Therapy

A number of novel strategies targeting TAMs properties (recruitment, polarization, survival, phagocytosis, angiogenesis) are currently in clinical development [17]. The inhibition of CSF1 (M-CSF) interaction with its receptor CSF1R, by small molecules or neutralizing anti-CSF1R (e.g., PLX3397, GW2580) or anti-CSF1 (e.g., emactuzumab, cabiralizumab) monoclonal antibodies (mAbs), showed promising antitumor efficacy by inhibiting M2-like phenotype of TAMs, increasing the number of infiltrating CD8^+^ T cells, improving the response to antiangiogenic (anti-VEGF), ICIs (anti-PD-1 and anti-CTLA4) and chemotherapy (i.e., paclitaxel) [17,110,111].

As mentioned, CCL2 mediates the most well characterized pathway that promotes the recruitment of myeloid cells expressing its receptor (CCR2) and, accordingly, CCR2 blockade inhibited infiltration and immunosuppressive polarization of TAMs and MDSCs, in preclinical models [112,113]. Several CCR2 inhibitors (CCX872-B, PF-04136309, MLN1202, and BMS-813160) are currently in clinical trials for the treatment of solid tumors, in combination with other therapeutic agents (i.e., FOLFIRINOX regimen), achieving at least partial response and local tumor control [17]. Another promising target for cancer therapy is IL-1β that promotes recruitment and proliferation of myeloid cells into TME [114]. The IL-1 receptor antagonist/IL-1Ra (anakinra) prevents myeloid cells’ accumulation and tumor progression in different mouse models and, in combination with other agents, is currently under Phase II clinical investigation [114].

Strategies to dampen M2-immunosuppressive signature in TAMs (re-polarization or reprogramming) are acquiring increasingly beneficial impact in cancer therapy [17]. In preclinical studies, genetic and pharmacological inhibition of phosphoinositide 3-kinase (PI3K)γ blocked the accumulation of immunosuppressive MDSCs in TME [115] and skewed TAMs to produce higher levels of proinflammatory cytokines (e.g., TNFα, IL-12), finally resulting in significant reduction in tumor progression [115,116]. The PI3Kγ inhibitor IPI-549 is currently in clinical testing in multicenter Phase Ib trials for several advanced solid tumors, in combination with anti-PD-1 nivolumab [17,117]. TMP195, a class IIa histone deacetylase (HDAC) inhibitor, has been reported to repolarize TAMs towards an M1-like phenotype and to synergize with PD-1 inhibitors, reducing tumor burden and metastasis in an autochthonous mouse model of breast cancer [118].

STAT3 is a key driver for immunosuppressive activity in both MDSCs and TAMs and its inhibition is showing promising therapeutic possibilities [119]. Indeed, inhibition of STAT3 produces a re-polarization of TAMs toward pro-inflammatory and anti-tumoral phenotype [120]. The STAT3 inhibitor TTI-101 is now undergoing testing in a Phase I clinical trial in patients with advanced cancers [17].

Activation of CD40, a member of the TNF receptor family, promotes production of proinflammatory factors by macrophages, including IL-1β, IL-12, TNFα, and NO [121]. CD40 agonists are reported to induce anti-tumoral cytotoxic activity by TAMs; several CD40-activating antibodies and recombinant ligands are currently under clinical trials for solid tumors as single agents or in combination with chemo- and immuno-therapy, or tumor vaccines [17].

Generally, complement cascade activation acts to boost the innate immune response. However, in cancer settings, TAMs upregulated the production of complement component C5a, which in turn binds its receptor (C5aR) on macrophages, resulting in M2-polarization and favouring CD8^+^ T cell inhibition and cancer progression [122]. The peptide antagonist PMX-53 blocking C5aR in combination with paclitaxel effectively inhibits tumor growth by repolarizing TAMs towards the proinflammatory phenotype [17]. The “don’t eat me” signal CD47 is a self-molecule expressed by cancer cells that protects them from phagocytosis by binding to the signal regulatory protein (SIRP)1α on macrophages [123]. Agents blocking CD47–SIRP1α interaction (e.g., TTI-621) promotes macrophage-mediated phagocytosis of cancer cells and T cell activation, thus reducing tumor growth [124].

#### 3.5.2. Targeting MDSCs in Cancer Therapy

Preclinical and clinical studies showed that treatment with tadalafil, an inhibitor of phosphodiesterase-5 (PDE5), significantly inhibited MDSCs’ suppressive functions by the downregulation of iNOS and ARG1, boosting tumor-specific immunity [17,125]. Accordingly, entinostat, a class I HDAC inhibitor, in combination with anti-PD-1 antibodies, reduces the expression of ARG1, iNOS, and COX2 in MDSCs, finally inhibiting their immunosuppressive capacity and delaying tumor growth in mice [126]. As for TAMs, STAT3 drives the immunosuppressive state of MDSCs [127]. STAT3 siRNA or decoy oligonucleotides have been used in combination with ICIs to improve therapeutic efficacy [128]. Selective delivery to MDSCs of STAT3 inhibitors, through their coupling with the TLR9 ligand CpG oligonucleotides, reduced the immunosuppressive activity of MDSCs [129].

Further, intensive investigations were performed to block the chemokine-dependent recruitment of MDSCs to TME. Blockade of CCL2/CCR2 axis reduced the frequency of immunosuppressive myeloid cells and showed anti-tumoral efficacies in different preclinical cancer models [130]. Inhibitions of CCR5 chemokine receptor, expressed on a specific subset of MDSCs, prevents their recruitment and immunosuppressive functions in cancer patients [131]. In addition, patients with a mutated CCR5 variant were reported to be resistant to prostate cancer development; this is corroborated by the evidence that CCR5^+^ MDSCs from melanoma patients displayed a more immunosuppressive pattern compared to the CCR5^−^ counterpart [132].

Other evidence reported that all-trans retinoic acid (ATRA), a vitamin A derivative binding to the retinoic acid receptor, alone or in combination with other agents (i.e., IL-2, DCs vaccine) decreased the frequency of immunosuppressive MDSCs, inducing their differentiation into mature DCs and/or macrophages and downregulating ROS production [133]. Interestingly, tumor-derived extracellular vesicles (TEV) were reported to promote differentiation of non-immunosuppressive immature myeloid cells into MDSCs and to activate their anti-inflammatory profile [134]. In line with this, inhibitors of TEV release (i.e., dimethylamiloride or omeprazole) reduced MDSCs’ expansion and immunosuppressive activities [135]. Receptor tyrosine kinase (RTK) inhibitors (such as sunitinib), which can block VEGF and c-kit signalling, decrease the number of circulating MDSCs, reducing STAT3 activation and ARG1 expression and increasing T cells’ activity and proliferation [136].

Several chemotherapeutic agents such as gemcitabine, 5-fluorouracil, docetaxel, doxorubicin and paclitaxel, in different mouse cancer models, reduced the frequency of MDSCs in favour of M1-like cells, enhancing the effector functions of T and NK cells and decreasing tumor burden [137].

## 4. Obesity, Lipid Metabolism, Inflammation and Immunosuppressive Myeloid Cells

In recent years, increasing evidence underlined a complex interplay between energy metabolism and immune cell responses. Indeed, the emerging research field of immunometabolism aims to elucidate the bi-directional causal relationship between metabolic reprogramming and immunological dysfunctions in various pathological conditions, such as metabolic syndrome, autoimmune diseases and cancer [138]. In this perspective, the alteration of energy metabolism into tumor microenvironment has been recently pointed to as a fuel of neoplastic cell proliferation, as well as an orchestrator of cancer-related inflammation and immune escape [139]. Of interest, systemic dysregulation of energy metabolism induced by obesity appears to support cancer-related immune dysfunctions [26]. Obesity is mainly characterized by an excess of white adipose tissue (WAT) whose primary function is to regulate systemic energy homeostasis by storing energy in form of lipids, largely as triglycerides (3 fatty acids combined to glycerol). The uncontrolled lipid storage and hyper-adiposity produce metabolic dysfunctions, such as insulin resistance and dyslipidaemias, namely abnormal amounts of blood triglycerides and cholesterol [140].

Compelling evidence indicates that obesity portends worse clinical outcomes after cancer diagnosis and supports several pathophysiological events associated with tumor progression [26]. These events include inflammation and oxidative stress, which modulate tumor-promoting cytokines and immune response [26,141]. Obesity generates a chronic low-grade inflammation or “metaflammation” (inflammation in metabolic tissues), characterized by a modest increase in inflammatory factors and absence of clinical signs (hence the term ‘subclinical inflammation’) [28,142], thus providing an intricate network of cytokines, adipokines and metabolites that affect different components of the immune system, including myeloid cells, and predisposing to cancer development [26]. In a simplistic description, the adipocyte-derived factors, such as free fatty acids and leptin, stimulate resident adipose tissue macrophages (ATM) to release cytokines (e.g., TNFα, IL-6) and directly tune hematopoiesis to further engage monocytes/macrophage and other immune cells (e.g., T cells) into adipose tissues [28]. The immune cell recruitment involves several inflammatory mediators, such as CCL2, S100A8/9, IL-1β, IL-5, GM-CSF and lipid-derived PGE2, which promote cancer-related inflammation, emergency myelopoiesis and tumor progression (Figure 2) [27,28,143].

In a parallel mode, the tumor microenvironment induces a metabolic reprogramming of both immune and cancer cells. Myeloid cells appear increasingly relevant at the crossroad of this intricate network, even due to their plasticity in reprogramming both energetic and inflammatory profile in response to microenvironmental stress.

### 4.1. Fatty Acids, Macrophage Polarization and MDSC Functions

The dichotomy of M1/M2 macrophages is reflected also by their distinctive metabolic traits. M1 polarization enhances aerobic glycolysis and pentose phosphate pathway (PPP), sustaining an anabolic phenotype, while M2 macrophages engage catabolic metabolism via oxidative phosphorylation (OXPHOS) [22]. Similarly, fatty acids’ metabolism is oriented toward their biosynthesis (FAS) in M1 macrophages, while M2 polarization activates β-oxidation (FAO), generating high levels of ATP and acetyl-CoA; this last participates in the Krebs cycle (TCA cycle) and cholesterol biosynthesis [25].

Distinct microenvironmental stimuli finely modulate the lipid metabolism of macrophages [144]. LPS-stimulated macrophages upregulate the expression of both the glucose importer GLUT1 and glycolytic genes to provide ATP in a faster way and induce ATP-citrate lyase (ACLY), which converts citrate from TCA cycle in acetyl-CoA necessary for lipid biosynthesis. Inhibition of glycolysis or ACLY activity reduces the production of inflammatory mediators, such as IL-1β, NO and ROS [145,146]. Acetyl-CoA serves as a building block for enzymatic activity of fatty acid synthase (FASN), whose expression is necessary for M1 activation. Indeed, LPS-challenged mice showed FASN activation via uncoupling protein 2 (UCP2) in macrophages, inducing NLRP3 inflammasome and production of both IL-1β and IL-18. UCP2 deficiency prevents pro-inflammatory signals and fatty acid synthesis [147]. The rate of fatty acid synthesis is controlled at transcriptional level by the sterol regulatory element-binding proteins (SREBPs); the isoform 1 (SREPB1a) is abundantly induced in LPS-stimulated macrophages, positively regulating NLRP3-dependent IL-1β secretion [148]. Lipidomic analysis showed that TLR4 agonists profoundly rearrange the asset of intracellular and plasma membrane lipids (eicosanoids, sphingolipids, sterols) in macrophages, through the activation of STAT3 and NF-κB pathways [149,150]. Beyond the intracellular production of fatty acids, the inflammatory activity of macrophages is also induced by exogenous lipids. In particular, circulating saturated fatty acids (e.g., palmitate), which are increased during obesity, can trigger the NLRP3 inflammasome activity in combination with LPS [151]. However, the specific mechanisms by which endogenous and exogenous fatty acids induce inflammasome activation in macrophages are not fully understood. A recent study reported that oxidized phospholipids, derived from 1-palmitoyl-2-arachidonyl-*sn*-glycero-3-phosphorylcholine (PAPC), potently boosts the production of IL-1β by rewiring the metabolism of LPS-stimulated macrophages to potentiate, along with glycolysis, also mitochondrial respiration and glutaminolysis [152].

In contrast to M1 macrophages, their IL-4- and STAT6-dependent M2 polarization downmodulates the glucose flux, while upregulates the lipoprotein lipase (LpL) and CD36 which both mediate fatty acid uptake and induced-expression of the peroxisome proliferator-activated receptor (PPAR)γ and its coactivator PGC1β, thus driving the metabolic switch towards mitochondrial respiration and FAO pathways [153,154]. The role of FAO in M2 polarization is controversial. IL-4-stimulated macrophages treated with etomoxir, inhibitor of carnitine palmitoyltransferase (CPT)1a, a rate-limiting enzyme of FAO that mediates the transport of fatty acids on outer mitochondrial membrane, showed a reduction in the M2 inflammatory phenotype, in both human and murine cells [155,156]. However, macrophage-specific deletion of the fatty acid importer CPT2, located on the inner mitochondrial membrane, precludes FAO without affecting the expression of M2 polarization markers [155]. More recently, it has been proposed that etomoxir inhibits M2 polarization by depleting the intracellular free Coenzyme A (CoA) pool, rather than antagonizing CPT1a or CPT2 [157].

The involvement of these mechanisms in the differentiation of immunosuppressive tumor-associated macrophages is currently not widely defined. However, cancer cells adapt and undergo metabolic changes acquiring a lipogenic phenotype and inducing a pro-tumorigenic lipid network [158]. Similar to tumor cells, TAMs alter their lipid profile [24]. In renal cell carcinoma (RCC), TAMs produce high levels of lipid-derived 15-hydroxyeicosatetraenoic acid through the enzymatic activity of 15-lipoxygenase 2 (15-LOX2), which positively correlates with CCL2 and IL-10 secretion, supporting pro-tumoral inflammation [159]. Intriguingly, the differential profiles in lipid-derived inflammatory mediators (e.g., prostaglandins, leukotrienes) reflect the macrophage heterogeneity during cancer development. In Lewis lung carcinoma, tissue-resident alveolar macrophages express COX1, 5-lipoxygenase (5-LOX) and high levels of leukotrienes; whereas bone marrow-derived TAMs upregulate COX2 and prostaglandins production, which positively correlate with increased tumor angiogenesis and MDSCs’ expansion [67,160,161]. Lewis lung carcinoma cell-derived M-CSF induced FAS activation in TAMs, leading to a pro-tumoral IL-10-producing phenotype and to the pro-angiogenic activation of nuclear receptor PPARβ/δ [162]. In a mouse mammary tumor model, the differential expression of epidermal fatty acid binding protein (E-FABP) discriminated two subsets of TAMs, with different inflammatory activities. FABPs orchestrated the biological functions of various lipophilic molecules, including long-chain fatty acids and eicosanoids [163]. E-FABP-expressing TAMs produced higher levels of IFNβ associated with increased intracellular lipid droplet formation, which correlates with a more M1-like phenotype that favors recruitment of anti-tumor NK cells [164].

Cancer-associated MDSCs shift their main source of energy from glycolysis toward FAO. This metabolic reprogramming is consistently more evident in tumor-infiltrating MDSCs as compared to circulating MDSCs, both in mouse models and human patients, and is characterized by increased CD36-mediated fatty acid uptake and higher expression of key enzymes (e.g., CPT1a, ACADM, HADHA, PGC1β), that in turn upregulate the rate of FAO necessary for the production of immunosuppressive ARG1 and cytokines driving MDSCs’ expansion (G-CSF, GM-CSF, IL-1β, IL-6 and IL-10) [165]. In addition, cancer cell-derived G- and GM-CSF act via STAT3/5 in a paracrine manner on infiltrating MDSCs, to upregulate the expression of CD36 receptor and enhance the uptake of exogenous fatty acids. STAT3 or STAT5 inhibition, or CD36 deletion, prevented lipid metabolism and the immunosuppressive functions of MDSCs [166].

Fatty acid transport proteins (FATPs), as long-chain fatty acid transporters, were shown to be upregulated in tumor-derived MDSCs and to control their suppressive activity [167,168]. In particular, in response to the GM-CSF/STAT5 signalling, FATP2 increased the uptake of fatty acids, including arachidonic acid needed for the production of PGE2, which boosts the immunosuppressive activity of PMN-MDSCs [168].

In contrast to the anti-inflammatory role of PPARγ in IL-4-polarized macrophages, the enhanced PPARγ signalling in cancer-associated PMN-MDSCs decreases the lysosomal acid lipase (LAL), generating reduced ROS production and impairing cancer cell proliferation and metastasis [169]. Additional evidence showed that polyunsaturated fatty acids (PUFA), such as linolenic acid (ω-3) and (ω-6), as well as a fish oil-based diet (rich in ω-3 PUFA), are able to induce accumulation of MDSCs both in vitro and in vivo [170]. Recent in vitro experiments showed that treatment of the myeloid suppressor cell line MSC-2 with sodium oleate and lineolate (unsaturated fatty acids), but not with stearate (saturated FA), increases intracellular lipid droplet formation paralleled by potent suppressive functions on activated T cells [171].

In addition to TAMs and MDSCs, tumor-associated DCs from mice and cancer patients are also influenced by intracellular lipid content. Tumor-derived or TLR-activated DCs increases FAS and intracellular triglycerides storage [172]. Lipid accumulation in tumor-associated DCs, mediated by upregulation of scavenger receptor A (SR-A1 or MSR1), negatively regulates their capacity to process antigen via MHC class II and to stimulate allogenic T cells [173,174].

Taken together, immunosuppressive myeloid cells rely on FAO as a major metabolic circuit sustaining the production of inhibitory cytokines. However, the specific mechanisms driving this metabolic shift in tumor microenvironment need further characterization, especially regarding the MDSC subsets.

### 4.2. Cholesterol Metabolism and Tumor-Associated Myeloid Cells

Cholesterol is an essential lipid constituent of cell membranes, precursor of steroid hormones, vitamin D and bile acids, and transporter of fatty acids in the form of cholesterol esters [175]. Cholesterol sources are both endogenous and exogenous. In both cases, the liver is central for its management. Exogenous cholesterol is obtained directly from the diet, absorbed by enterocytes and, packaged into chylomicrons, reaches the liver via lymphatic and blood circulation [176]. The endogenous form is synthetized, starting from acetyl-CoA, through several enzymatic steps [175]. The key enzyme HMGCR (target of statins) is subjected to feedback inhibition by cholesterol itself at the transcriptional level through the SREBPs system [175]. Due to its low water solubility, cholesterol is transported into the bloodstream together with triglycerides, complexed with apolipoproteins (e.g., ApoA1, ApoB, ApoC2, ApoE) to form chylomicrons, VLDL, IDL, LDL, HDL, mediating lipid delivering to tissues [177]. Excesses of intracellular cholesterol and its derivatives (i.e., oxysterols) are directly sensed by LXRα/β nuclear receptors, which induce the transcription of genes involved in reverse cholesterol transport (RCT). Among these, membrane transporters ATP-binding cassette (ABC)A1 and ABCG1 mediate the efflux of cholesterol to ApoA1, forming high-density lipoprotein (HDL), which carries cholesterol and oxysterols back to the liver for their re-use or intestinal excretion through bile acids [178]. The uptake of low-density lipoprotein (LDL) cholesterol via the LDL-receptor (LDLR) is regulated by negative feedback to avoid intracellular cholesterol overload. However, macrophages are equipped with several scavenger receptors, which mediate the clearance of excessive oxidized (ox)LDL. Cholesterol-laden macrophage (foam cells) formation is a crucial step in the pathogenesis of atherosclerosis [179]. In industrialized societies, the major cause of hypercholesterolemia (high LDL- and low HDL-cholesterol in blood) is the excessive consumption of high-calorie diets, known as Western-type diets (WDs) [178,180].

Besides cardiovascular diseases (CVDs), some epidemiological evidence associated high serum cholesterol levels with a higher risk and a poor prognosis in different cancers [181,182,183]. Conversely, other studies suggested opposite conclusions, showing that statins (cholesterol-lowering drugs) and/or low cholesterol levels had no effect or even higher risk for bladder, colon and lung cancer [183,184,185]. In spite of these epidemiologic controversies, preclinical studies more consistently suggested a pro-tumoral role of cholesterol, with a higher impact addressed to deregulated intracellular cholesterol in cancer cells, than circulating cholesterol [183]. Overall, although hypercholesterolemia may have a relevant role during oncogenesis and tumor development, these discrepant observations suggest that other levels of interaction should be considered, such as the tissue of cancer origin and/or the influence of immune response.

The impact of myeloid cells on cholesterol metabolism is poorly investigated and controversial. The cholesterol handling capacities by M1 and M2 macrophages were examined in vitro. Some evidence showed that IL-4-treated macrophages upregulate the expression of CD36 and macrophage scavenger receptor 1 (MSR1) and uptake more lipids and cholesterol than LPS/IFNγ-stimulated macrophages, despite no difference in ApoA1 expression or in HDL-induced cholesterol excretion [186]. Conversely, the influence of cholesterol metabolism alterations on myeloid cell functions is more widely characterized, especially in the context of cardiovascular diseases and atherosclerosis [178,187]. Indeed, excess LDL is absorbed into the intima of arterial wall, generating inflammatory signals (e.g., IL-6, TNFα, IL-1β) and tuning hematopoietic output and myeloid cell recruitment, in particular Ly6C^hi^ monocytes. Once infiltrated into arterial tissue, monocytes differentiate into macrophages, scavenge oxLDL cholesterol and become foam cells. Cholesterol-loaded macrophages exert dynamic inflammatory patterns ranging between M1 and M2 phenotype, at various stages and in different tissue sites along the pathological process [179,188].

Hypercholesterolemia also promotes hematopoietic stem cells (HSCs) proliferation and release of granulocyte-macrophage progenitors (GMP) in circulation, as well as myelo/monocytic differentiation, resulting in a higher frequency of monocytes and neutrophils. In addition, elevated LDL cholesterol induces increased CXCL12 plasma level, promoting mobilization of CXCR4^+^ bone marrow progenitor cells. These effects appear to be counteracted by HDL [189,190].

Oxysterols, biologically active cholesterol derivatives produced enzymatically or by oxidative stress, are highly represented in obesity and hypercholesterolemia [191]. Although certain oxysterols showed anti-tumoral effects in a cancer cell-intrinsic fashion [192], others exert dual pro-tumoral and anti-inflammatory properties. Indeed, in different breast cancer models, it was described that the 27-hydroxycholesterol (27-OHC), produced by the CYP27A1 enzyme, binds and interacts with both LXRβ and the estrogen-receptor (ER)-α and consequently promotes tumor progression, not only by direct effects on proliferating cancer cells but also by inducing high rates of HSC mobilization and differentiation of immunosuppressive granulocytes [193,194,195]. However, a discrepant clinical observation underlined that higher circulating 27-OHC was associated with lower risk of breast cancer in postmenopausal women [196]. The expression of CH25H enzyme and its product 25-hydroxycholesterol (25-OHC) are remarkably induced in macrophages and dendritic cells by different stimuli owing either pro- or anti-inflammatory activity. 25-OHC has a role downstream of viral-induced type I IFNs in reducing IL-1β production. In contrast, LPS or TLR3 agonists upregulate 25-OHC, increasing macrophage production of pro-inflammatory cytokines, such as IL-6, IL-8 and M-CSF [197]. Human monocytes treated in vitro with constant levels of oxysterols (i.e., 27-hydroxycholesterol) acquire a prominent polarization toward an M2 immunomodulatory phenotype [198]. In a tumor setting, certain oxysterols (i.e., 22- and 24-OHC) promoted immunosuppression by enhancing the recruitment of pro-tumoral neutrophils in a CXCR2-dependent manner [199]. Further, 25- and 27-OHC, as well as cholesterol precursors and other oxysterols, interact with LXRs. A number of pieces of evidence correlate LXR activation with anti-inflammatory phenotype in macrophages and dendritic cells under different inflammatory conditions, including cancer [200]. Indeed, 22-OHC and 25-OHC produced by tumor cells impair antigen presentation by DCs through the inhibition of CCR7 expression, in an LXR-dependent manner [201]. In addition, tumor-derived oxidized cholesterol esters decreased the cell surface expression of peptide-MHC class I complexes and blocked the tumor antigen cross-presentation activity of DCs [174]. Contrarily, a recent study underlined a key role of LXRs activation in the promotion of anti-tumoral immune response. Pharmacological LXRβ agonism reduced MDSCs’ frequency in association with enhanced activation of cytotoxic T lymphocytes (CTLs), in both murine models and in cancer patients. It has been also proposed that increased expression of ApoE by LXR agonists induces the apoptosis of MDSCs interacting with the low-density lipoprotein receptor-related protein 8 (LRP8) [202].

Mice deficient in ABC transporters (ABCA1/ABCG1) showed hyper-proliferation of hematopoietic progenitor cells, resulting in monocytosis and neutrophilia, and activation of pro-inflammatory gene expression, suggesting an anti-inflammatory role mediated by cholesterol efflux [203,204]. However, the role of reverse cholesterol transport in cancer-related immune response is controversial. ApoA1 and HDL were reported to exert an anti-tumoral role in B16 melanoma model by skewing TAMs towards an M1 phenotype [205]. Conversely, ABCG1 deletion is associated with an anti-tumoral M1 phenotype, increased NF-κB activation and inhibition of tumor growth in high-cholesterol diet-fed mice [206]. In line with these findings, the block of cholesterol efflux in myeloid cells, by deletion of ABCA1 gene or ABCA1 and ABCG1 genes, resulted in a significant reduction in tumor growth, associated with reduction in MDSCs [207]. Accordingly, a recent study showed that membrane cholesterol efflux of macrophages via ABCA1/G1 enhances their pro-tumoral activation in response to IL-4 [208,209].

Recent evidence demonstrated that obesity in both mice and human increases the frequency of MDSCs [210,211], an observation confirmed in different cancer models (breast, kidney, melanoma, prostate). Although some molecular mechanisms have been defined to drive obesity/MDSCs’ regulation (e.g., IL-5, GM-CSF, CCL2, leptin), the role of metabolic components influencing this scenario is poorly clarified [210,212,213,214,215]. In a recent characterization of granulocytic subsets in cancer patients, it was unravelled that the scavenger receptor for oxidized LDL (LOX-1) is a specific marker discriminating the immunosuppressive PMN-MDSC subset from mature and pro-inflammatory neutrophils [216]. Noteworthy, PD-1 is also expressed on myeloid cells, inducing accumulation of GMP and MDSCs and, thus, cancer progression. PD-1 deletion in myeloid cells increases cholesterol biosynthesis in response to myeloid growth factors (i.e., G-, GM-CSF) and enhances T and myeloid cell differentiation and function, promoting innate and adaptive antitumoral response [217].

In addition to the above-mentioned controversy on statins’ effect in cancer, the inhibition of cholesterol biosynthesis showed conflicting inflammatory activity. Most of the studies highlighted a potent impact of statins treatment in the modulation of pro-inflammatory mediators produced by macrophages and in promoting MDSCs’ expansion during cancer, as well as inflammatory diseases and atherosclerosis [218]. However, other evidence showed that statins reduced TAMs’ frequency and their M2 polarization in human lung adenocarcinoma, despite any beneficial effect in patients’ outcome [219]. A recent in vitro characterization of statins effect on human monocyte/macrophage inflammatory phenotype showed that cytokine production of freshly isolated and LPS-stimulated monocytes is not altered by statins. In contrast, the pro-inflammatory response of overnight-differentiated cells after LPS treatment is drastically reduced by statins, suggesting their effect in keeping cells in a “monocyte-like” (activable) state [220]. The epigenetic reprogramming of innate immune cells has been recently linked to a phenomenon known as ‘trained immunity’ or innate immune memory. Emerging evidence indicates that not only pathogens, but also hypercholesterolemia and ox-LDL, are capable of inducing trained immunity via epigenetic modifications of monocytes and that this imprinting persists after treatment with statins despite normalization of circulating cholesterol [221,222].

In sum, the metabolic competition and crosstalk between cancer cells and immune cells is a crucial determinant for metabolic alterations that influences pro- or anti-inflammatory functions of myeloid cells.

## 5. Targeting Lipid Metabolism in Myeloid Cells to Improve Cancer Immunotherapy

Cancer immunotherapy has drastically increased the survival rate of cancer patients as compared to chemo- or radio-therapy treatments. However, to date, the benefits have been limited to a minority of patients and cancer types. In addition, a majority of patients treated with immunotherapy are either primary non-responders or eventually develop immune-refractory progressive disease and require additional therapy [223]. An improved understanding of molecular mechanisms driving immune cells to participate in resistance to immunotherapy could extend clinical benefit to the majority of patients. In this regard, evidence has shown that TAMs and MDSCs may play a critical role in hindering immunotherapy efficacy, creating a systemic immunosuppressive status that impairs effector T cell activation [68,224]. Since metabolic reprogramming drives immune cell development and function, use of metabolism-targeting drugs could offer new opportunities in strengthening cancer immunotherapy. To date, the role of fatty acid and cholesterol metabolisms appear more clearly involved in T and NK cells, as a tool for improving immunotherapy [225,226,227,228,229]. However, emerging evidence is highlighting the reprogramming of lipid metabolism in myeloid cells as an effective approach to restrain the immunosuppressive TME.

### 5.1. Targeting Fatty Acid Metabolism

Evidence in Lewis lung (LLC) and CT26 colon cancers showed that tumor-derived factors increase the uptake of fatty acids via FATP2 and the subsequent PGE2 release by PMN-MDSCs, which directly correlated with CD8^+^ T cell suppression. Of note, administration of lipofermata, a FATP2 inhibitor, reduced tumor progression in different cancer models. In addition, combined treatments of lipofermata with ICIs (anti-CTLA4 antibody) potently inhibited LLC progression [168]. Combination with anti-PD-1 produced similar effects, although with a lesser extent, in the TC-1 lung tumor model. The beneficial effects of these combination therapies were ascribed to a reduced release of PGE2 by PMN-MDSCs and increased infiltration and activation of CD8^+^ T cells. Importantly, in combination with antibody-blocking CSF1R, lipofermata produced anti-tumor effects in the LLC model, suggesting a possible involvement of lipid uptake by TAMs, since anti-CSF1R alone did not produce beneficial effects [168,230]. However, this hypothesis needs further elucidation.

Similarly, adoptive cell transfer (ACT) in combination with etomoxir, an inhibitor of the rate-limiting enzyme in the FAO cycle, CTP1, drastically reduced tumor progression in the LLC model, as compared to ACT or etomoxir treatment alone. This effect was accompanied by increased infiltration of adoptively transferred T cells in the TME and increased production of IFNγ [165]. Interestingly, etomoxir did not affect MDSCs’ frequency, whereas it reduced ARG1 expression, ROS/RNS release, as well as cytokines involved in MDSC expansion (i.e., G-CSF, GM-CSF, IL-6, IL-10). Of note, FAO inhibition significantly increases antitumor potency of low-dose chemotherapy by targeting MDSC-associated immune suppression [165]. In addition, etomoxir treatment prevents overexpression of PGC1β and M2 macrophage polarization, potentiating their pro-inflammatory response [156,157].

Biguanides, such as metformin and phenformin, are a class of drugs commonly used in the treatment of type II diabetes, capable of inhibiting complex I of the mitochondrial respiratory chain, a key complex in electron transfer during fatty acid oxidation [231]. Beyond an intrinsic and not completely understood anticancer activity of biguanides, phenformin treatment revealed diminished spleen and tumor infiltration of PMN-MDSCs in the BP01 melanoma model [232]. Moreover, phenformin, in combination with anti-PD-1 administration, reduced the immunosuppressive expression of ARG1 by MDSCs, resulting in a synergistic effect in the induction of CD8^+^ T cells’ infiltration in TME and reduced tumor growth [232].

Different pre-clinical and clinical evidence also showed that metformin administration inhibited MDSCs’ accumulation and immunosuppressive functions through several mechanisms, such as reduction in STAT3 phosphorylation, ARG1 and ROS production [233], decrease in CXCL1 secretion by tumor cells [234] and downregulation of ectonucleotidases CD39/CD73 catalyzing adenosine production, which confers immunosuppressive functions to MDSCs [235]. Moreover, metformin increased cytotoxic T cell activity by degrading the PD-L1 protein expression by cancer cells [236].

PPARγ is a nuclear receptor that regulates lipid uptake and intracellular metabolism. As described above, PPARγ overexpression regulates anti-inflammatory cytokines production by M2-polarized macrophages [153]. By contrast, in MDSCs, PPARγ activation dampens immunosuppression and tumor progression [169]. In agreement, mice harboring a selective deficiency of PPARγ in Lysozyme-2-expressing cells (e.g., macrophages) showed impaired tumor rejection after treatment with GM-CSF-secreting and irradiated tumor cell-based vaccine (GVAX) [237]. Conversely, administration of PPARγ agonist rosiglitazone, an approved drug for the treatment of type II diabetes, augmented CD8^+^ T cytotoxicity in a myeloid cell-dependent manner and increased tumor destruction in combination with CTLA4 antibody blockade and GVAX [237]. Furthermore, in a pancreatic cancer model, the combination of gemcitabine and rosiglitazone diminished MDSC-induced immunosuppression of CD8^+^ T cells [238].

Interesting lipid metabolism targets may reside in the COX2/mPGES1/PGE2 axis. Indeed, the overexpression of COX2 and microsomal prostaglandin E synthase-1 (mPGES1) by MDSCs and TAMs promotes the upregulation of arachidonic acid towards PGE2 conversion, directly associated with increased expression of PD-L1 by MDSCs and TAMs [239]. Administration of the selective COX2 inhibitor, celecoxib, in combination with anti-PD-1, inhibited PD-L1 expression by myeloid cells and strongly reduced cancer progression in both the B16-F10 melanoma and 4T1 breast cancer models [239,240].

There is evidence that a substantial proportion of DCs in both tumor models and cancer patients have increased amounts of lipids, specifically triglycerides, due to increased synthesis of fatty acids or increased lipid uptake from plasma via MSR1 [173]. These lipid-laden DCs are immature and have a profound defect in their ability to process and present soluble antigens. Using either 5-(tetradecycloxy)-2-furoic acid (TOFA), an inhibitor of acetyl-CoA carboxylase (ACC) that participates in fatty acid synthesis, or neutralizing antibodies to MSR1 before the incubation of DCs with tumor supernatant, substantially enhanced the anti-tumor potency of DCs vaccination [173].

Taken together, this evidence suggests that MDSC and TAM fatty acid metabolism might be targeted to amplify the efficacy of cancer immunotherapy.

### 5.2. Targeting Cholesterol-Related Pathways

Despite the fact that few studies are available on the effects elicited by cholesterol metabolism and immunotherapy on MDSC and TAM functions, emerging evidence shows a potential benefit of targeting de novo cholesterol synthesis as well as reverse cholesterol transport in different immune cell types and cancers [241]. It has been previously reported that LXR activation via ApoE induction inhibits melanoma cell proliferation, inducing apoptosis, reduced cancer angiogenesis and metastatic progression [242,243]. More recently, it was found that the LXR agonist (RGX-104) induces regression of several preclinical cancers, including ovarian cancer, melanoma and glioblastoma, through the reduction in tumor-infiltrated MDSCs’ frequency. In particular, Tavazoie et al. showed that co-administration of LXR agonists with adoptive transfer of CTLs bearing transgenic T cell receptors specific for the melanoma tumor antigen gp100 (pmel), into vaccinated B16F10 tumors bearing mice, enhanced both the anti-tumor activity of transferred CTLs and mouse survival [202]. Similarly, administration of RGX-104, in combination with anti-PD-1, significantly enhanced anti-tumor activity in B16F10 melanoma and LLC models, as compared to single agent anti-PD1 therapy. The triple combination therapy, comprising adoptive T cells mounting the gp100-specific TCR, anti-PD-1 and LXR agonist, remarkably enhanced anti-tumor activity and was well tolerated [202]. In addition, the combination of LXR agonist with anti-PD-1 significantly reduced MDSCs and tumor growth in a melanoma model resistant to anti-PD-1 therapy [202]. Accordingly, avasimibe, an inhibitor of the cholesterol esterification enzyme ACAT1, with a good safety profile in humans, significantly empowered the anti-cancer response of CD8^+^ T cells in melanoma-bearing mice, despite no differences in MDSCs’ frequency [226]. Further, the combination of avasimibe with anti-PD-1 antibody had a better efficacy than monotherapies in melanoma and LLC models, due to the drastic potentiation of effector functions in both PD-1^hi^ and PD-1^lo^ CD8^+^ T cells [226].

In a recent study, treatment with the TLR9 agonist CpG increased mitochondrial abundance and FAO in TAMs, while diverting acetyl-CoA towards the biosynthesis of cholesterol. This event reduced tumor progression, endowing TAMs with a higher phagocytic capacity, bypassing the CD47 immune checkpoint [244].

Intriguingly, a dual relationship between cholesterol biosynthesis and STING-dependent type I IFN response was observed in macrophages and dendritic cells. While IFNβ signaling decreases cholesterol biosynthesis, reduced intracellular cholesterol availability drives type I IFN responses and potentiate pro-inflammatory signals, such as IL-1β and CXCL10 [245]. This suggests that disruption of cholesterol biosynthesis could enhance the beneficial effect of STING agonists in cancer immunotherapy. In line with this hypothesis, statins treatment exerted an anti-tumor activity, synergizing with IL-2 administration in NK cell activation, in a myeloid cell-dependent manner [246]. In particular, statins plus IL-2 increased IL-1β and IL-18 production in CD56^+^ DC/macrophage-like cells, which in turn cooperate with IL-2 to induce IFNγ release by NK cells, dampening cancer progression [246]. This evidence suggests the critical role of cholesterol metabolism in the phenotypic adaptation of myeloid cells in neoplastic diseases.

## 6. Conclusions

Immunotherapy is providing unprecedented success and new opportunities for treatment of cancer patients. In spite of this, a large part of patients undergoing immunotherapy display unresponsiveness or severe side effects, highlighting the need for new studies that might improve our understanding of the mechanisms guiding these therapeutic limitations. Tumor microenvironmental conditions, including microphysiological conditions, metabolic and immune profiles, dictate the reprogramming and the functional setting of infiltrating immune cells, which in turn influence the response to therapy.

Myeloid cells represent the majority portion of the immune tumor infiltrate, which orchestrates the multidirectional interplay between metabolic pathways and immune responses within the TME. New evidence indicates lipid metabolism among the main determinants of tumor progression. In particular, the reprogramming of lipid metabolism in TAMs and MDSCs appears to contribute significantly to the development of their pro-tumoral phenotype, while emerging evidence suggests that metabolic drugs that refine fatty acids and cholesterol metabolism may offer new opportunities to reshape myeloid cell functions and improve the effectiveness of therapy. Furthermore, with an increasing proportion of the overweight or obese population and the paradoxical beneficial effect of obesity in patients treated with anti-PD1 [225], the effect of obesity and dyslipidemias on cancer immunotherapy must be dissected and clarified.

## Figures and Tables

**Figure 1 ijms-21-05845-f001:**
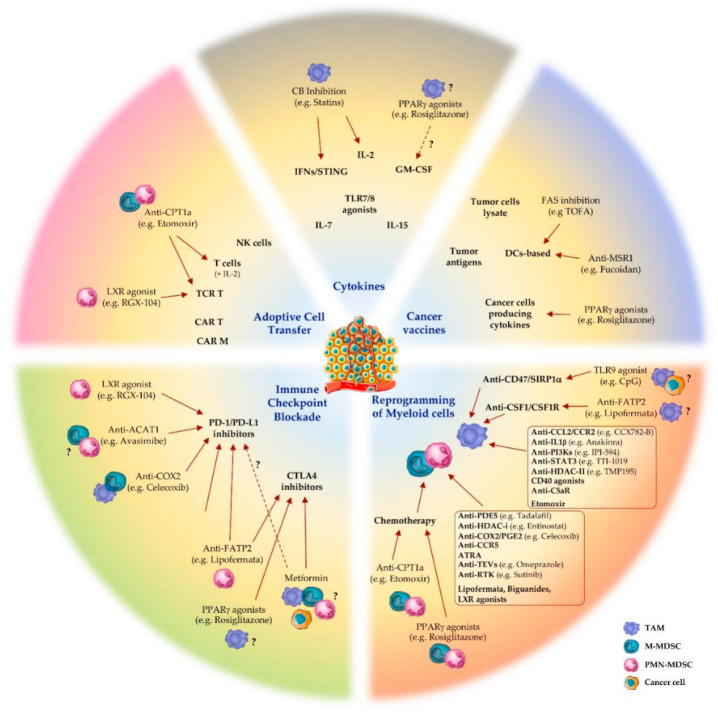
Targeting lipid metabolism of myeloid cells in cancer immunotherapy. Pharmacological modulation of molecular regulators of lipid metabolism in tumor-associated macrophages (TAM), monocytic (M-) and granulocytic (PMN-) myeloid-derived suppressor cells (MDSC) showed efficacy in dampening tumor immunosuppression and improving cancer immunotherapies: immune checkpoint blockade; adoptive cell transfer; cytokines administration or modulation; cancer vaccines; reprogramming of polarized inflammation in myeloid cells. *Abbreviations:* LXR, liver X receptor; ACAT1, acetyl-CoA acetyltransferase 1; COX2, cyclooxygenase 2; FATP2, fatty acid transport protein 1; PPARγ, peroxisome proliferator-activated receptor γ; PD-1, programmed-death protein 1; PD-L1, PD-1 ligand; CTLA4, cytotoxic T-lymphocyte antigen 4; CPT1a, carnitine palmitoyltransferase 1a; CAR T/M, chimeric antigen receptor T cell/Macrophage; TCR T, T cell receptor-engineered T cell; NK, natural killer; CB, cholesterol biosynthesis; IFN, interferon; STING, stimulator of interferon genes; TLR, Toll-like receptor; GM-CSF, granulocyte-macrophage colony-stimulator factor; FAS, fatty acid synthesis; MSR1, macrophage scavenger receptor 1; DC, dendritic cell; SIRP1α, signal regulatory protein 1α; CSF1, macrophage colony-stimulating factor; CSF1R, CSF1 receptor; PI3K, phosphoinositide 3-kinase; HDAC, histone deacetylases; C5aR, complement component 5 receptor; PDE5, phosphodiesterase 5; COX2, cyclooxygenase 2; PGE2, prostaglandin E2; CCR5, chemokine receptor 5; ATRA, all-trans retinoic acid; TEV, tumor-derived extracellular vesicles; RTK, receptor tyrosine kinase. Text in brackets represents examples of drugs targeting the molecule or pathway of reference. Dashed arrows and question marks underline controversial or not fully clarified evidence.

**Figure 2 ijms-21-05845-f002:**
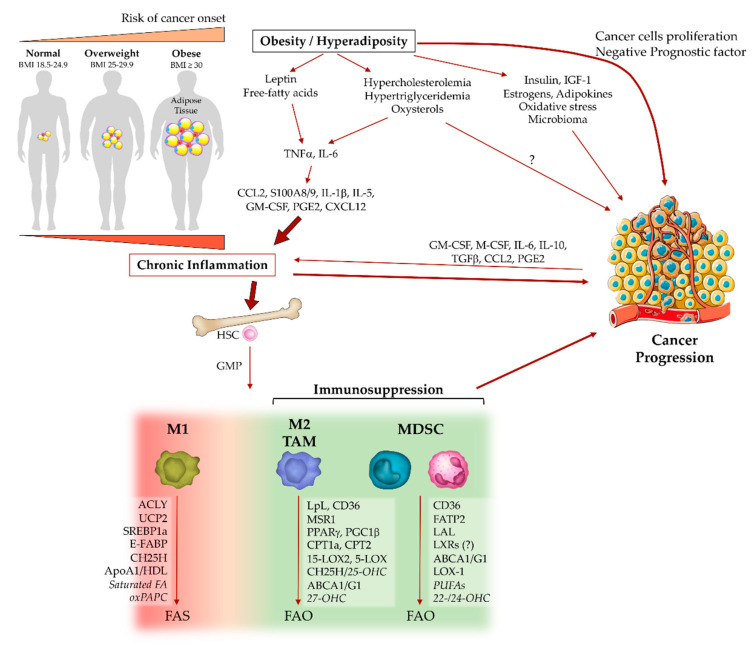
Influence of obesity on cancer-associated inflammation, suppressor myeloid cells and cancer progression. Obesity represents a risk factor for many cancers. Weight gain generates an excess of adipose tissue depots associated with dyslipidemias (i.e., hyper-cholesterolemia and -triglyceridemia) and metabolic alterations (e.g., hyperinsulinemia), which can promote tumor progression. In addition, obesity produces a chronic inflammatory state, which primes myelopoiesis and the immunosuppressive phenotype of tumor-associated myeloid cells. Reprogramming of intracellular lipid metabolism also occurs in myeloid cells from normal-weight cancer bearers in response to TME-derived factors. Several molecular players are associated with either pro- or anti-inflammatory (M1/M2) states of macrophage activation, as exemplified by tumor-associated macrophages (TAM) and myeloid-derived suppressor cells (MDSC). M1 macrophage polarization activates fatty acid synthesis (FAS), whereas immunosuppressive M2/TAMs and MDSCs shift toward fatty acid oxidation (FAO). *Abbreviations:* BMI, body mass index; IGF-1, insulin-like growth factor 1; HSC, hematopoietic stem cell; GMP, granulocyte-macrophage progenitor; ACLY, ATP-citrate lyase; UCP2, uncoupling protein 2; SREBP, sterol regulatory element-binding protein; E-FABP, epidermal fatty acids binding protein; CH25H, cholesterol 25-hydroxylase; ApoA1, apolipoprotein A1; HDL, high-density lipoprotein; FA, fatty acid; oxPAPC, oxidized 1-palmitoyl-2-arachidonyl-*sn*-glycero-3-phosphorylcholine; LpL, lipoprotein lipase; MSR1, macrophage scavenger receptor 1; PPARγ, peroxisome proliferator-activated receptor γ; PGC1β, PPARγ coactivator 1β; CPT1a/2, carnitine palmitoyltransferase 1a/2; 15-LOX2, 15-lipoxygenase 2; 5-LOX, 5-lipoxygenase; ABCA1/G1, ATP-binding cassette A1/G1; 22-/24-/25-/27-OHC, 22-/24-/25-/27-hydroxycholesterol; FATP2, fatty acid transport protein 1; LAL, lysosomal acid lipase; LXR, liver X receptor; LOX-1, oxidized low-density lipoprotein receptor 1; PUFA, polyunsaturated fatty acid. Question marks underline controversial or not fully clarified evidence. Words in italics represent lipidic metabolites.

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
