# Peer review of "Lipid Metabolism and Cancer Immunotherapy: Immunosuppressive Myeloid Cells at the Crossroad"

_ijms, 2020, doi:10.3390/ijms21165845_

Round 1
Reviewer 1 Report
The manuscript called „Lipid metabolism and cancer immunotherapy: 2 immunosuppressive myeloid cells at the crossroad“ describes the relationship between lipid metabolism and immune system. The topic is really highly relevant and up-to-date.
The manuscript is written and designed well. The main weekness of this manuscript are abbreviations. There is a huge of abbreviations without previous explanation, some of them are explained later in the text, some of the are not explained at all. Please, check all of the abbreviations. Please, provide also a list of abbreviations. The manuscript contains few grammatical and stylistical errors.
The figures are well, however, they contain also many abbreviations without the later explanations. All of the abbreviations have to be explained in the figure legend.
After minor revisions, I would recommend the manuscript to be accepted.
Author Response
We thank the Referee for his appreciation and commenting that our review covers a timely hot topic, and for his suggestion to accept the manuscript after minor revisions. We have implemented our review accordingly to the referee comments.
- We accepted the suggestion of the Referee to revise and re-organize the abbreviations in the text and figures. We added the explanation of relevant abbreviations in the text at their first mention.
We added a complete description of abbreviations in figure legends. Additionally, as suggested by the Reviewer, we added a list of the abbreviations frequently cited in the text (pag. 18) (text highlighted in green);
- We revised the grammar and the linguistic style of text, modifying sentences, words and expositions where appropriate, and without affecting the meaning and content of the text.
Reviewer 2 Report
This is an excellent review that elegantly gives a comprehensive overview of immunometabolism in tumor immune suppression by myeloid cells and therapeutic responses of cancer patients. The first part is an informative description of myeloid cells and immunotherapy. The second part describes the most relevant findings of lipid metabolism effects on TAM and MDSC function in tumors and highlighted the therapeutic potential of targeting fatty acids and cholesterol metabolism as new approach of anticancer immunotherapy.
The review is well written, describes a timely hot topic and summarized the most relevant recent finding from preclinical and clinical reports.
For improvement, authors might reduce the first part-describing tumor associated myeloid cells and cancer immunotherapy in order to keep light-shed on the original and novel part related to lipid and tumor immunity.
Author Response
We thank the Referee for the general appreciation that our review is well constructed and describes a highly relevant and up-to-date topic. We accepted the suggestion to shorten the first part of the manuscript in order to emphasize the part concerning lipid metabolism in myeloid cells and its relevance in anticancer immunotherapy. In particular, we reduced more extensively the Chapter 2 (from n=134 lines to n=105 lines) trying to preserve the most relevant notions regarding phenotype, origin and function of TAMs and MDSCs. Minor reductions, instead, were done in chapter 3.
We highlighted in yellow the more extensively modified paragraphs.
Reviewer 3 Report
This is a very interesting review. While the part related to lipds is satisfactory, the one related to cholesterol is not that strong.
Specifically the relationship between cholesterol and cancer is still a matter of debate an must be exposed with caution. If statin were reported to decrease patient's death (ref 183), that does not mean that hypercholesterolemia is a risk factor for canncers. Invers correlation bewteen The relationship betaween cholesterol, oxysterol and cancer are not that simple as reported in ref 184. this is a growing domin in which several cholesterol metabolites were shown to display various and apparent contradictory properties. ex : 27Hydroxy cholesterol was shown to display tumor promoter in breast cancer and immunosuppressive properties in mice. In that case researchers suggested that hyper cholesterolemia and high levels in 27HC (which is the major circulating metabolite in human) was involved in breast cancer recurrence. In an other hand results from the EPIC-Heidelberg clinical trial suggest that 27HC is predictive of patient survival (Lu et al, JNCI, 2019).
The word "oxysterols" is refering to oxygenation products of cholesterol that display very different properties and targeting according their structures. most oxysterols have no impact at all on immunity and cancers. Some such as 6-oxocholestan-diol, have been identified as tumor promoters activating glucocorticoid and liver-X-receptors, while others such as dendrogenin A have been shown to display tumor suppressor properties through the regulation of Liver-X-receptor-beta. This reveals the complexity of the role of cholesterol in cancers and suggests the existence of a balance between tumor promoter and tumor suppressors derived from cholesterol in tissues. The fact that these metabolites are targeting LXR and GR suggest also that they can display immunosuppressive or immunostimulating properties through these targets.
There is no evidences that hypecholesterolemia is associated to a higher risk of cancers.
thus to indicate oxysterols on figure 2 as vector of immunosuppression is a mistake.
paragraph 4.2: "targeting cholesterol metabolism" this title is too general and must be restricted to the topics of papers they are discussed here.
Author Response
We thank the Referee for the general appreciation that our review is interesting. We thoughtfully considered his helpful comments and suggestions and we tried our best to make clearer specific contradictory or not optimally exposed concepts. We highlighted in light blue the modification inserted in the text of manuscript.
Please, see here below the point-to-point reply (in black, Reviewer comments; in blue, our replies).
- This is a very interesting review. While the part related to lipds is satisfactory, the one related to cholesterol is not that strong.
- The aim of our manuscript is a comprehensive overview of the cross-interaction between lipid metabolism and inflammatory profile of myeloid cells in cancer and in anticancer immunotherapy; the relation between lipids and cancer progression and the intrinsic lipid pathways in cancer cells are not the main focus of our review. Therefore, while the evidences regarding fatty acids in myeloid cells are more extensive and better clarified in literature, less is known about cholesterol-related pathways in myeloid cells especially in tumor settings and cancer immunotherapy, as we mentioned in the text (lines 733-734).
- Specifically the relationship between cholesterol and cancer is still a matter of debate an must be exposed with caution. If statin were reported to decrease patient's death (ref 183), that does not mean that hypercholesterolemia is a risk factor for canncers. Invers correlation bewteen The relationship betaween cholesterol, oxysterol and cancer are not that simple as reported in ref 184 (Ref. 182 in the revised manuscript, Ding X, 2019)
- We are in agreement with the Reviewer about the controversial role of cholesterol in cancer and we did not want to assert that hypercholesterolemia is a substantial risk factor for cancers. Indeed, in lines 558-559 and 563 we underlined the controversies of epidemiological and pre-clinical evidences regarding high serum cholesterol levels and cancer occurrence, as well reported in the cited review article from Ref. 183 (Kuzu, 2016). To fulfill any misunderstanding, in the revised manuscript we modified and specified with a more cautious language the exposure of this misleading concept (lines 556-557, 560).
- this is a growing domin in which several cholesterol metabolites were shown to display various and apparent contradictory properties. ex : 27Hydroxy cholesterol was shown to display tumor promoter in breast cancer and immunosuppressive properties in mice. In that case researchers suggested that hyper cholesterolemia and high levels in 27HC (which is the major circulating metabolite in human) was involved in breast cancer recurrence. In an other hand results from the EPIC-Heidelberg clinical trial suggest that 27HC is predictive of patient survival (Lu et al, JNCI, 2019)
- As already reported in lines 584-591, we described the pro-tumoral activity of 27-hydroxycholesterol (27-OHC) focusing preferentially on its immune-related role more than its cancer cell-intrinsic activity. As suggested by the Reviewer, we specified its contradictory role in the prediction of patient survival citing the reference paper suggested (Ref. 197, Lu et al, JNCI, 2019) (lines 591-593), despite any specific correlation with the myeloid and immune cells were assessed in this study.
- The word "oxysterols" is refering to oxygenation products of cholesterol that display very different properties and targeting according their structures. most oxysterols have no impact at all on immunity and cancers. Some such as 6-oxocholestan-diol, have been identified as tumor promoters activating glucocorticoid and liver-X-receptors, while others such as dendrogenin A have been shown to display tumor suppressor properties through the regulation of Liver-X-receptor-beta. This reveals the complexity of the role of cholesterol in cancers and suggests the existence of a balance between tumor promoter and tumor suppressors derived from cholesterol in tissues. The fact that these metabolites are targeting LXR and GR suggest also that they can display immunosuppressive or immunostimulating properties through these targets
- We agree with criticisms described by the Reviewer in regard of oxysterols activity in cancer; thus, in the revised manuscript, we specified more clearly the evidence of their opposing role in cancer cells (lines 585-586, Ref. 192). However, our main aim was to describe the capacity of oxysterols to modulate the activity of tumor-associated myeloid cells. We accepted the suggestion to not generalize the concept that all oxysterols exert this property. Therefore, we underlined more precisely the specific oxysterols identified for their anti-inflammatory and/or immunosuppressive activity in tumor-associated myeloid cells (lines 600, 604 and Figure 2).
Further, we described the contradictory role of LXRs activity in tumor-associated myeloid cells (lines 602-613) associated (Ref. 201) or not (Ref. 202) to specific evidences of its oxysterols-mediated activation.
- There is no evidences that hypecholesterolemia is associated to a higher risk of cancers
- See point 2
- thus to indicate oxysterols on figure 2 as vector of immunosuppression is a mistake
- See points 4.
- paragraph 4.2: "targeting cholesterol metabolism" this title is too general and must be restricted to the topics of papers they are discussed here.
- We accepted the suggestion of the Reviewer and we changed the paragraph title from "targeting cholesterol metabolism" to “Targeting cholesterol-related pathways”